# Preclinical Evaluation of a Novel Small Molecule Inhibitor of LIM Kinases (LIMK) CEL_Amide in Philadelphia-Chromosome Positive (*BCR::ABL+*) Acute Lymphoblastic Leukemia (ALL)

**DOI:** 10.3390/jcm11226761

**Published:** 2022-11-15

**Authors:** Jeannig Berrou, Mélanie Dupont, Hanane Djamai, Emilie Adicéam, Véronique Parietti, Anna Kaci, Emmanuelle Clappier, Jean-Michel Cayuela, André Baruchel, Fabrice Paublant, Renaud Prudent, Jacques Ghysdael, Claude Gardin, Hervé Dombret, Thorsten Braun

**Affiliations:** 1Laboratoire de Transfert des Leucémies, URP-3518, Institut de Recherche Saint-Louis, Université Paris Cité, 75010 Paris, France; 2INSERM/CNRS, US53/UAR2030, Institut de Recherche Saint-Louis, Université Paris Cité, 75010 Paris, France; 3Laboratory of Hematology, Hôpital Saint-Louis (Assistance Publique–Hôpitaux de Paris and Université Paris Cité), 75010 Paris, France; 4Department of Pediatric Hemato-Immunology, Hôpital Universitaire Robert Debré (Assistance Publique–Hôpitaux de Paris and Université Paris Cité), 75010 Paris, France; 5CELLIPSE Company, 38000 Grenoble, France; 6CNRS UMR3348, INSERM U1278, Institut Curie, Centre Universitaire Bat 110, 91405 Orsay, France; 7Hematology Department, Hôpital Avicenne (Assistance Publique–Hôpitaux de Paris and Université Paris XIII), 93000 Bobigny, France; 8Leukemia Unit, Hematology Department, Hôpital Saint-Louis (Assistance Publique–Hôpitaux de Paris and Université Paris Cité), 75010 Paris, France

**Keywords:** LIM kinase, CEL_Amide, *BCR::ABL*, ALL, tyrosine kinase inhibitors

## Abstract

Ph+ (*BCR::ABL+*) B-ALL was considered to be high risk, but recent advances in *BCR::ABL*-targeting TKIs has shown improved outcomes in combination with backbone chemotherapy. Nevertheless, new treatment strategies are needed, including approaches without chemotherapy for elderly patients. LIMK1/2 acts downstream from various signaling pathways, which modifies cytoskeleton dynamics via phosphorylation of cofilin. Upstream of LIMK1/2, ROCK is constitutively activated by *BCR::ABL*, and upon activation, ROCK leads to the phosphorylation of LIMK1/2, resulting in the inactivation of cofilin by its phosphorylation and subsequently abrogating its apoptosis-promoting activity. Here, we demonstrate the anti-leukemic effects of a novel LIMK1/2 inhibitor (LIMKi) CEL_Amide in vitro and in vivo for *BCR::ABL*-driven B-ALL. The IC50 value of CEL_Amide was ≤1000 nM in *BCR::ABL+* TOM-1 and BV-173 cells and induced dose-dependent apoptosis and cell cycle arrest in these cell lines. LIMK1/2 were expressed in *BCR::ABL+* cell lines and patient cells and LIMKi treatment decreased LIMK1 protein expression, whereas LIMK2 expression was unaffected. As expected, CEL_Amide exposure caused specific activating downstream dephosphorylation of cofilin in cell lines and primary cells. Combination experiments with CEL_Amide and *BCR::ABL* TKIs imatinib, dasatinib, nilotinib, and ponatinib were synergistic for the treatment of both TOM-1 and BV-173 cells. CDKN2A^ko^/*BCR::ABL1*+ B-ALL cells were transplanted in mice, which were treated with combinations of CEL_Amide and nilotinib or ponatinib, which significantly prolonged their survival. Altogether, the LIMKi CEL_Amide yields activity in Ph+ ALL models when combined with *BCR::ABL*-targeting TKIs, showing promising synergy that warrants further investigation.

## 1. Introduction

Acute lymphoblastic leukemia (ALL) results from leukemic transformation of lymphoid B- and T-lineage precursors caused by multiple genetic events, including chromosomal aberrations, gene mutations, and rearrangements [1]. In contrast to other subtypes of ALL, including the more frequent pediatric patients, the incidence of the Philadelphia chromosome (Ph+)-positive B-ALL (*BCR::ABL+*) increases with age in up to 50% of patients and was considered to signal a poor prognosis before tyrosine kinase inhibitors (TKI) targeting *BCR::ABL* were combined with classical backbone chemotherapy [2,3]. Even strategies without classical chemotherapy have been tested using combinations of TKI and bispecific and conjugated antibodies [4,5]. Nevertheless, relapsed and refractory disease remains frequent in the elderly and new treatment strategies are needed.

The inhibition of kinases in the signaling pathway acting downstream of *BCR::ABL+* would be of particular interest for relapsed and resistant disease caused by mutations of the *BCR::ABL* kinase domain and subsequent clonal evolution [6,7]. It was demonstrated that Rho kinases (ROCK) were constitutively activated in *BCR::ABL*-transformed cells in a way that was dependent on the *BCR::ABL*-induced activation of PI3 kinase and RhoGTPase [8]. As a result of pharmacologic inhibition of ROCK, *BCR::ABL*-mediated aberrant signaling was disrupted and inhibited cell growth of *BCR::ABL+* cell lines, even those bearing the T315I mutation that confers resistance to all available TKIs, except ponatinib [8]. Nevertheless, no ROCK inhibitors have entered clinical trials so far.

LIMK1 and LIMK2 are two isoforms of serine/threonine kinases and their activation is caused by phosphorylation at threonine 508 by ROCK [9,10]. The downstream targets of LIMK1 and LIMK2 are members of factors of depolymerisation of the actin–(ADF) family called cofilin [11]. In its inactivated state, cofilin is phosphorylated at serine 3 by LIMK1 or LIMK2, and dephosphorylation caused by inhibiting LIMK1/2 leads to the activation of cofilin, and consequently to actin reorganization and mitochondrial apoptosis in cancer cells [12,13]. The precise mechanisms that links LIMK inhibition to apoptosis are not yet fully elucidated. The inhibition of LIMK leads indirectly to mitochondrial apoptosis, as activated LIMK inactivates cofilin by phosphorylation. As a result of the inactivation of LIMK, cofilin is activated by dephosphorylation, which causes mitochondrial translocation of cofilin, leading to mitochondrial apoptosis characterized by cytochrome *c* release and subsequent activation of caspase-3 [13,14]. For pediatric t(8;21) *RUNX1-RUNX1T1* AML, it was shown that overexpression of the Ras homolog family member (RHOB) correlated with resistance to chemotherapy and relapse [15]. RHOB re-organizes the actin cytoskeleton by activating the ROCK-LIMK-cofilin pathway, which increases blast adhesion with stress fiber formation. This leads to reduced mitochondrial apoptotic cell death after exposure to chemotherapeutic agents. Furthermore, LIMK inhibition leads to centrosome fragmentation caused by the disruption of proper kinetochore–microtubule interactions at the onset of anaphase [16]. This leads to impaired Aurora-A kinase activation, resulting in inhibition of cell growth and apoptosis in leukemic cells.

ROCK phosphorylation is also triggered by *FLT3-ITD* mutations in AML. Based on this rationale, we recently reported preclinical activity of a highly specific small molecule LIMK1 and LIMK2 inhibitor (LIMKi) CEL_Amide in *FLT3-ITD+* cell lines and in a xenograft mouse model [17]. LIMKi had antiproliferative effects in *FLT3-ITD+* cells and decreased constitutive phosphorylation of cofilin. LIMKi yielded synergy with different *FLT3-ITD+* inhibitors in vitro and in vivo. Notably, exposure to CEL-Amide had no significant toxicity for CD34+ cells obtained from healthy donors, which is encouraging for future potential of clinical testing [17].

Here, we show preclinical activity of LIMKi CEL_Amide in different *BCR::ABL+* models: CEL_Amide had antiproliferative effects in *BCR::ABL+* cells and in vivo activity in a CDKN2A^ko^/*BCR::ABL1+* mouse model that abrogated the inactivating phosphorylation of cofilin. Furthermore, CEL_Amide synergized with commercially available TKIs and backbone chemotherapy, delineating a novel treatment strategy for *BCR::ABL+* ALL.

## 2. Materials and Methods

### 2.1. Compounds

CEL_Amide (632.1 g·mol^−1^ as sodium salt) was donated from Cellipse. It was resuspended in water as a 5 mM stock solution and stored at −20 °C. Imatinib was diluted in DMSO as a 6 mM stock solution. Dasatinib and nilotinib were diluted in DMSO as a 50 mM stock solution. Vincristin was resuspended in DMSO as a 100 mM stock solution. Dexamethasone and ponatinib were resuspended in DMSO as a 150 mM stock solution. The drugs were obtained from Clinisciences (Nanterre, France), except ponatinib, which was obtained from Euromedex (Souffelweyersheim, France); all were stored at −20 °C.

### 2.2. Cell Lines and Primary Patient Cells

BV-173 and TOM-1 were purchased from the Deutsche Sammlung für Mikroorganismen und Zellkulturen (DSMZ, Braunschweig, Germany). Cell lines were grown in RPMI 1640 (Life Technologies, Courtaboeuf, France), supplemented with 20% heat-inactivated fetal calf serum, 2 mM L-glutamine, 100 IU/mL penicillin, and 100 µg/mL streptomycin (Dominique Dutscher, Brumath, France), at 37 °C with 5% CO_2_. Patients and healthy donors provided informed consent prior to bone marrow aspiration or blood sample collection, following the Declaration of Helsinki. MNC were obtained after gradient centrifugation with a separation medium for lymphocytes (Eurobio, Courtaboeuf, France).

### 2.3. Quantitative-Real Time Polymerase Chain Reaction (RT-qPCR)

Extraction of total RNA was performed with Maxwell^®^ RSC simplyRNA kit (Promega, Charbonnières Les Bains, France). Then, RNA was titrated with the NanoDrop 2000 c UV-Vis spectrophotometer (Thermo Fisher Scientific, Courtaboeuf, France) and frozen at −80 °C. Complementary DNA (cDNA) was synthesized from 500 ng RNA with a reverse transcriptase PrimeScriptTM kit (TaKaRa, Ozyme, Saint-Cyr-l’École, France). RT-qPCR was performed for LIMK1 and LIMK2 in 10 µL from one-tenth of the cDNA volume (10 ng RNA), using a thermocycler StepOnePlusTM (Thermo Fisher Scientific, Courtaboeuf, France) with SYBR^®^ Green reagent (Thermo Fisher Scientific, Courtaboeuf, France) in standard mode (1 cycle of 30 s at 95 °C, followed by 40 cycles of 5 s at 95 °C, then 30 s at 60 °C), with a supplementary melting curve step for SYBR^®^ Green assays. Primers (Eurogentec, Angers, France) are listed in Appendix A. GAPDH served as a control gene for normalization of mRNA levels.

### 2.4. Immunoblotting

A total of 5 × 10^6^ cells were used for protein extraction. An amount of 30 μg of protein was loaded on SDS-polyacrylamide gels using 4–15% gradient gels (Bio-Rad, Marnes-La-Coquette, France). After migration, proteins were transferred to nitrocellulose membranes using a Mini Trans-Blot Electrophoretic Transfer Cell (Bio-Rad, Marnes-La-Coquette, France). Membranes were blocked with a blocking buffer (TBS (Tris Buffer solution) 1× with 5% BSA), and incubated with the primary antibody overnight at 4 °C in the same buffer: anti-LIMK1 (#3842S, Ozyme, Saint-Cyr-l’École, France), anti-LIMK2 (#3845S, Ozyme, Saint-Cyr-l’École, France), anti-cofilin (#5175, Ozyme, Saint-Cyr-l’École, France), anti-phosphocofilin (#3313, Ozyme, Saint-Cyr-l’École, France), or anti-GAPDH (#398600, Life Technologies, Courtaboeuf, France). Secondary antibodies were goat anti-rabbit or goat anti-mouse HRP conjugate (#1706515 or #1706516, Biorad, Marnes-La-Coquette, France), incubated for 1 h at room temperature in TBS 1X, 5% BSA, revealed with an enhanced chemiluminescence detection system (ECL, Biorad, Marnes-La-Coquette, France) and visualized on the membrane with the Chemidoc^TM^ Touch Imaging System (Biorad, Marnes-La-Coquette, France).

### 2.5. MTS Assay, Apoptosis Assessment, and Cell Cycle Analysis

For the MTS assay, cells were incubated in 96-well plates at 20,000 cells per well, and treated with a broad range of CEL_Amide concentrations (78 nM–10 μM) for 72 h. An amount of 20 µL per well of MTS solution (Promega, Charbonnières Les Bains, France) were placed into each well and incubated in the dark at 37 °C for 4 h. Absorbance was read at 490 nm using GloMax Explorer (Promega, Charbonnières Les Bains, France). Three independent experiments were run for each cell line, and untreated cells were used as negative controls. The half maximal inhibitory concentration (IC50) values were calculated with Prism^®^ v6 software (GraphPad Software LLC, La Jolla, CA, USA).

For cell cycle analysis, 1 × 10^6^ cells were treated with 500 or 1500 nM of CEL_Amide for 48 h, then harvested, washed in PBS, and fixed in 70% ice cold ethanol. Cells were incubated with 100 µg/mL RNAse (Sigma, Saint Quentin Fallavier, France) and stained with 3 µg/mL propidium iodide (Becton Dickinson, Le Pont de Claix, France).

For apoptosis analysis, 1 × 10^6^ cell lines were resuspended in 1 mL culture medium and treated with a range of CEL_Amide concentrations 250–1500 nM or with CEL_Amide 1500 nM, nilotinib 25 nM, or ponatinib 1 nM for 72 h. Cells were stained with 1 µg/mL PI and Annexin-V-FITC (Becton Dickinson, Le Pont de Claix, France), according to the manufacturer’s instructions for 15 min at room temperature. Apoptotic cells were defined as Annexin V+ with or without PI uptake.

Cell cycle distribution and apoptosis were determined by cytofluorometric analysis using a CytoFLEX flow cytometer (Beckman Coulter, Villepinte, France) and analyzed with the FlowJo flow cytometry software (FlowJo LLC, Ashland, OR, USA).

### 2.6. Drug-Dose-Response Experiments and Combenefit Analysis

CEL_Amide was combined with different agents, including the TKIs imatinib, dasatinib, nilotinib, ponatinib, and vincristin or dexamethasone. Drugs were added simultaneously to cultured cell lines and relative cell numbers were measured at 72 h with MTS assays. IC50 values were computed with Prism^®^ v6 software (GraphPad Software LLC, La Jolla, CA, USA). The dose-response matrix was calculated with the Combenefit Software v.2.021 (Cancer Research UK Cambridge Institute, Cambridge, UK) by using the Loewe independence model [18].

### 2.7. Mouse Model

This study was carried out in accordance with the EC Directive 86/609/EEC for animal experiments and was approved by the Committee for Experimental Animal Studies of the University of Paris Institute Board Ethics. Animals were housed and bred at our animal facility (Institut de Recherche Saint-Louis, Saint Louis Hospital, Paris, France).

We used for 4 experimental groups this animal study, with a total of 30 female C57BL/6J mice. They were treated with vehicle (group 1; *n* = 5); CEL_Amide alone, administered ip, at 50 mg/kg (200 µL) twice a day (group 2; *n* = 5); nilotinib 80 mg/kg per os (400 µL) or ponatinib 5 mg/kg per os alone (400 µL) (group 3; *n* = 10); or a combination of CEL_Amide and nilotinib or ponatinib (group 4; *n* = 10). On day 0, 1 × 10^5^
*BCR::ABL*-induced B-ALL cells in 200 μL PBS were intravenously injected. Leukemic cells were generated by transduction of CDKN2A in deficient B-cell progenitors with a retrovirus coding for *BCR::ABL1* (P185) and GFP. This was followed by transplantation of these cells into sub-lethally-irradiated recipient C57BL/6J mice.

On day 15, nilotinib or ponatinib treatment was stopped. On day 25, CEL_Amide treatment was stopped. On days 8, 15, 22, and 29, a blood test was performed to measure the GFP percentage by flow cytometry analysis using a CytoFLEX flow cytometer (Beckman Coulter, Villepinte, France) and analyzed with the FlowJo flow cytometry software v.10 (FlowJo LLC, Ashland, OR, USA).

## 3. Results

### 3.1. Biological Effects of LIMKi in BCR::ABL+ Cell Lines

As *BCR::ABL+* activates the ROCK-LIMK axis, we sought to investigate the sensibility of *BCR::ABL+* B ALL to LIMK inhibition [8]. Thus, we determined effects of CEL_Amide in the *BCR::ABL+* B-ALL cell lines TOM-1 and BV-173. Cell viability after CEL_Amide exposure to increasing doses was measured with MTS assay in the cell lines. Significant inhibition of cell growth was found in TOM-1 (580 nM) and BV-173 (1090 nM) (Figure 1A). We also found dose-dependent apoptosis induction upon exposure to increasing doses of CEL_Amide (250–1500 nM) as detected by phosphatidylserine exposure and PI uptake up to 45% (TOM-1) and 50% (BV-173) at 48 h (Figure 1B). Furthermore, CEL_Amide caused cell cycle arrest in the G1/S transition (Figure 1C).

### 3.2. LIMK1/2 Are Expressed in BCR::ABL+ Cell Lines and CEL_Amide Inhibits Constitutive Phosphorylation of Cofilin

Furthermore, we measured mRNA expression of the two isoforms LIMK1 and LIMK2 in BV-173 and TOM-1 cell lines. LIMK1 and LIMK2 mRNA were expressed at the same level in the 2 *BCR::ABL+* cell lines (Figure 2A). We also found LIMK1 and LIMK2 protein expression in BV-173 and TOM-1 cell lines (Figure 2B). Upon treatment with CEL_Amide, we observed a dose-dependent decrease in LIMK1 protein expression in both tested *BCR::ABL+* cell lines. As inhibition of LIMK1/2 results in activating dephosphorylation of cofilin, we determined total and phosphorylated protein levels of cofilin (Figure 2B). As expected, we observed a significant decrease in the phosphorylated form of cofilin in *BCR::ABL+* cell lines, whereas total protein of cofilin was not modified. We also compared CD34+ enriched cells from three healthy donors with bone marrow blasts derived from four *BCR::ABL+* patients and found protein expression of LIMK1 and phosphorylated cofilin in *BCR::ABL+* blast cells, indicating constitutively increased cofilin inactivating activity of LIMK1 in patient-derived cells (Figure 2C). Patient characteristics are summarized in Appendix A.

### 3.3. BCR::ABL-Targeting TKIs Synergize with CEL_Amide for Treatment of BCR::ABL+ ALL Cell Lines

As *BCR::ABL*-targeting TKIs, used with chemotherapy and corticosteroids, are the standard of care of *BCR::ABL+* B-ALL, we were interested to test combinations of CEL_Amide with commercially available TKIs, including imatinib, dasatinib, nilotinib, and ponatinib, in BV-173 and TOM-1 cell lines. First, we tested the impact of nilotinib and ponatinib on protein expression of LIMK1 and phospho-cofilin and both TKIs had no effect on protein levels, indicating an absence of off-target effects in combination experiments with CEL_Amide (Appendix A). We found significant synergies of all TKIs tested in combination with CEL_Amide for both *BCR::ABL+* cell lines (Figure 3A). Furthermore, we tested indicated fixed doses for nilotinib and ponatinib and CEL_Amide and we observed significant increases in apoptosis for combinations in BV-173 cells from 40–50% up to 60–70% with CEL_Amide and nilotinib, and up to 80–90% for CEL_Amide with ponatinib (Figure 3B). Combination treatment of CEL_Amide with nilotinib or ponatinib enhanced cell cycle arrest in the G1/S transition in BV-173 and TOM-1 cell lines (Appendix A). Interestingly, we also found significant synergies for combination treatments of BV-173 and TOM-1 cell lines by CEL_Amide with either vincristin or dexamethasone (Appendix A).

### 3.4. Impact of Combined LIMKi and TKI Treatment on Survival in a BCR::ABL+ Mouse Model

To explore efficacy of the combination of CEL_Amide and *BCR::ABL*-targeting TKIs in vivo, we next sought to explore the therapeutic potential in a *BCR::ABL+* mouse model. C57Bl/6 mice were transplanted with GFP+ CDKN2A^ko^/*BCR::ABL1*+ B-ALL cells. Mice were treated with CEL_Amide alone, nilotinib, ponatinib, or a combination of LIMKi and TKI, compared with vehicle-treated mice (Figure 4A). Untreated mice died from this aggressive leukemia within 11–12 days (Figure 4B,C). CEL_Amide monotherapy had no impact on survival, whereas nilotinib and ponatinib monotherapy significantly extended mouse survival to 17 and 28 days, respectively (*p* = 0.0003 and *p* = 0.0005). Interestingly, combination therapy of CEL_Amide + nilotinib extended survival to 33 days (*p* = 0.0003) and CEL_Amide + ponatinib to 50 days (*p* = 0.0005). Leukemic GFP+ cells were scored once weekly in peripheral blood by flow cytometry and mice were considered leukemic if >1% GFP+ cells were detected in peripheral blood. CEL_Amide combined with nilotinib or ponatinib significantly delayed detection of GFP+ *BCR::ABL+* leukemic cells in peripheral blood (Appendix A).

## 4. Discussion

In our study, we report the anti-leukemic effects of CEL_Amide, a novel highly specific small molecule inhibitor of LIMK1/2, in *BCR::ABL+* ALL cells and in a *BCR::ABL+* mouse model, which, to our knowledge, is the first report of preclinical activity of a LIMKi in *BCR::ABL+* B-ALL. Interestingly, CEL_Amide significantly synergized with commercially available TKIs in vitro and in vivo. The outcomes of *BCR::ABL+* B-ALL have dramatically improved in recent years due to the introduction of *BCR::ABL*-targeting TKIs and antibodies, including the bispecific monoclonal antibody blinatumomab and the conjugated antibody inotuzumab ozogamicin, either frontline, in combination with classical chemotherapy, or in relapsed and refractory disease [1]. Nevertheless, 30–40% of patients are relapsing or are non-responders, especially elderly patients who are ineligible for allogenic hematopoietic stem cell transplantation [1,19,20]. Thus, the combination of LIMKi and TKI warrants further investigation, as this treatment strategy is synergistic in vitro and in vivo preclinical models where LIMKi act as a sensitizer to apoptosis. This strategy may potentially overcome resistance to *BCR::ABL*-targeting TKIs, which can emerge after point mutations of the breakpoint cluster region-ABL due to prolonged exposure to those drugs [6,7,21]. This hypothesis is supported by the observation that inhibition of ROCK or LIMK hyper-stabilizes mitotic spindles and impairs Aurora kinase A activation, suggesting the implication of the ROCK-LIMK-cofilin axis in Aurora kinase A activation via actin dynamics [16]. It was recently shown that Aurora kinase A is constitutively activated in *BCR::ABL+* cells [22]. Furthermore, knockdown of Aurora kinase A enhanced activity of *BCR::ABL*-targeting TKIs, including leukemic cells harboring *BCR::ABL* mutations that conferred resistance to TKIs. Thus, the synergy of LIMKi and *BCR::ABL+*-targeting TKIs may be due to indirect Aurora kinase A inhibition, and constitutes a suitable target to overcome TKI resistance.

In accordance with data obtained for ROCK inhibitors, which is a kinase that activates LIMK1 and 2 by phosphorylation, we could also show that LIMK1/2 inhibition had antiproliferative effects in *BCR::ABL+* cells [8]. Thus, targeting the ROCK-LIMK-cofilin axis constitutes a promising target in *BCR::ABL+* ALL.

It was shown that LIMK inhibition could sensitize cells to cell death induction, and here, we showed that CEL_Amide was synergistic in combination with *BCR::ABL+*-targeting TKIs [17]. It is possible that LIMKi by CEL_Amide monotherapy was not sufficient, as it was shown to only have modest activity in *FLT3-ITD+* AML [17]. Nevertheless, CEL_Amide monotherapy had significant biological effects, including strong apoptosis induction at higher doses in *BCR::ABL+* cells, but it is not known what plasma levels for CEL_Amide may be achievable in vivo. Thus, combination strategies may be the future direction for developing LIMKi, such as CEL_Amide and others, in cancer treatment [10].

We found that the two isoforms, LIMK1 and 2, were expressed in *BCR::ABL+* ALL cell lines and primary patient cells, whereas no significant expression of LIMK isoforms could be detected in CD34+ cells from healthy donors, underpinning their potential role in leukemic transformation. Upon treatment of BV-173 and TOM-1 cell lines with CEL_Amide, we observed degradation of LIMK1, whereas the LIMK2 protein was left unaffected, suggesting a possible role of LIMK1 in leukemic maintenance in *BCR::ABL+* ALL. Nevertheless, the precise mechanisms of LIMK-mediated leukemogenesis are not clearly elucidated and it was recently reported that upregulation of either LIMK1 or LIMK2 was observed to be dependent on the cellular system for AML cells [23]. This suggests that loss of one family member can be partially compensated for by the other, as a potential mechanism of resistance, encouraging the development of LIMK1/2 inhibitors to ensure inhibition of downstream signaling [23].

LIMK1 and LIMK2 phosphorylate cofilin, which leads to its inactivation and subsequent modulation and reorganization of the cytoskeleton, leading to the abrogation of mitochondrial apoptosis [13,24]. Here, we demonstrated that LIMK inhibition by CEL_Amide led to dephosphorylation of cofilin in *BCR::ABL+* ALL cell lines and patient cells. In turn, no significant amount of phospho-cofilin could be detected in cells derived from healthy donors. These findings underline the on-target effects of LIMKi. In future clinical trials, monitoring of phospho-cofilin in patient-derived cells could serve as a potential biomarker for efficient LIMK inhibition in vivo.

## 5. Conclusions

In conclusion, we report for the first time that the small molecule LIMK1/2 inhibitor CEL_Amide has anti leukemic activity in *BCR::ABL+* cells, leading to activation of cofilin and subsequent apoptosis and cell cycle arrest. Furthermore, combinations of CEL_Amide with *BCR::ABL*-targeting TKIs yield synergy in vitro and in vivo, encouraging further preclinical and clinical testing of LIMK1/2 inhibitors in *BCR::ABL+* B-ALL.

## Figures and Tables

**Figure 1 jcm-11-06761-f001:**
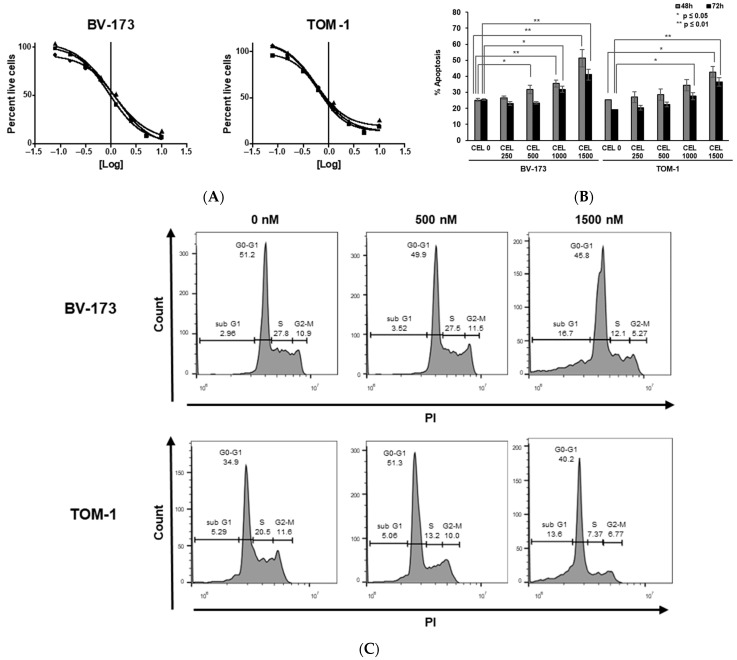
In vitro effect of LIMKi on viability, apoptosis, and cell cycle in *BCR::ABL1*+ ALL cell lines. IC50 was determined by MTS assay after exposure to increasing doses of CEL_Amide (78–10,000 nM) in indicated cell lines at 72 h. Results are shown as mean ± SEM from triplicate (*n* = 3) (**A**). Apoptosis of BV-173 and TOM-1 cell lines after 48 and 72 h treatment with different doses of CEL_Amide (250–1500 nM). Apoptotic cells were defined as Annexin V+ with or without PI uptake. Results are shown as mean ± SEM from duplicate (*n* = 3) (**B**). Cell cycle modifications at 48 h induced by increasing CEL_Amide doses (500–1500 nM) in BV-173 and TOM-1 cell lines (*n* = 3) (**C**).

**Figure 2 jcm-11-06761-f002:**
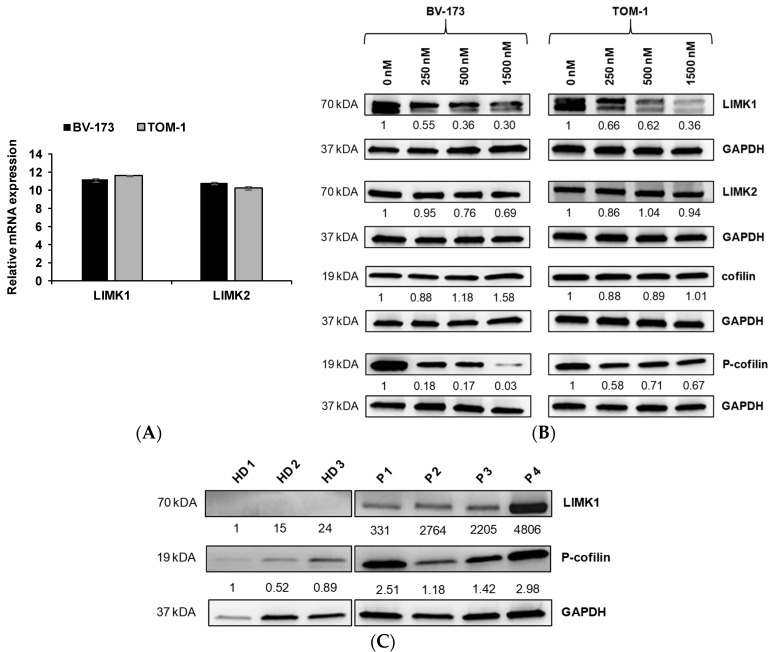
LIMK1 and LIMK2 basal expression and in vitro effect of LIMKi on LIMK1, LIMK2, cofilin, and phospho-cofilin protein expression in *BCR::ABL1*+ ALL cell lines and patient cells. LIMK1 and LIMK2 basal gene expression in BV-173 and TOM-1 cell lines were determined by RT-qPCR. Results are shown as mean ± SEM from triplicates (*n* = 3) (**A**). Western blot showing protein changes of LIMK1, LIMK2, cofilin, and phospho-cofilin in BV-173 and TOM-1 cell lines after 72 h exposure to increasing CEL_Amide doses (250–1500 nM). GAPDH was used as a loading control. One representative experiment out of three is shown (**B**). Western blot showing basal LIMK1 and phospho-cofilin expression in healthy donors (*n* = 3) and *BCR::ABL+* ALL patients (*n* = 4) ex vivo (**C**). Densitometry for LIMK1, LIMK2, cofilin, and phospho-cofilin proteins were reported as a ratio with GAPDH.

**Figure 3 jcm-11-06761-f003:**
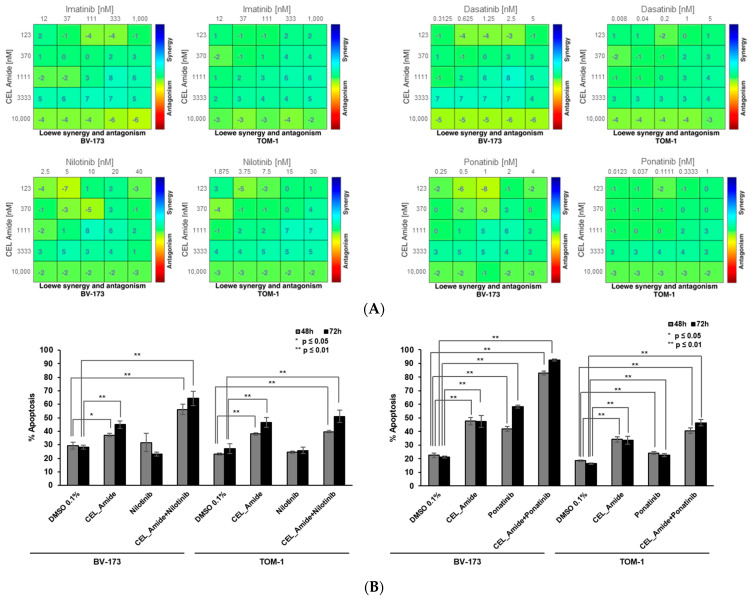
Drug combinations of LIMKi with *BCR::ABL*-targeting TKI inhibitors in *BCR::ABL1*+ ALL cell lines. BV-173 and TOM-1 cell lines were exposed for 72 h to increasing doses of different *BCR::ABL* inhibitors, including imatinib, dasatinib, and nilotinib or ponatinib, in combination with LIMKi CEL_Amide. The dose-response matrix was made according to the Loewe model. Results are shown from duplicates of 3 independent experiments (**A**). Apoptosis was measured in BV-173 and TOM-1 cell lines after 48 and 72 h exposure to CEL_Amide (1500 nM), nilotinib (25 nM), and ponatinib (1 nM), alone or in combination. Results are shown as mean ± SEM from duplicates (*n* = 3) (**B**).

**Figure 4 jcm-11-06761-f004:**
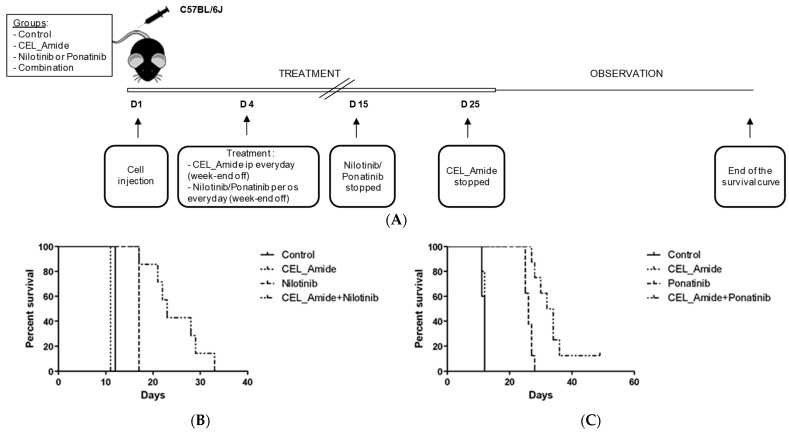
Effects of LIMKi in vivo. To evaluate LIMK1/2 inhibition by CEL_Amide in vivo, C57BL/6J mice were engrafted with GFP+ CDKN2Ako/*BCR::ABL1*+ B-ALL mice cells. Mice were treated with vehicle (group 1; *n* = 5); CEL_Amide alone (group 2; *n* = 5); nilotinib or ponatinib (group 3; *n* = 10); or a combination of CEL_Amide and nilotinib or ponatinib (group 4; *n* = 10) (**A**). Kaplan–Meyer survival curves for mice according to treatment combination of CEL_Amide with nilotinib (**B**). Kaplan–Meyer survival curves for mice according to treatment combination of CEL_Amide with ponatinib (**C**).

## Data Availability

Not applicable.

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
