# Peer review of "Preclinical Evaluation of a Novel Small Molecule Inhibitor of LIM Kinases (LIMK) CEL_Amide in Philadelphia-Chromosome Positive (BCR::ABL+) Acute Lymphoblastic Leukemia (ALL)"

_jcm, 2022, doi:10.3390/jcm11226761_

Round 1

Reviewer 1 Report

In this paper, the Authors report on the synergistic effect of LIMK inhibitors with TKIs in Philadelphia positive B cell ALL.

They show in a preclinical in vitro and in vivo model that the single agent treatment with LIMK inhibitor shows only modest activity, whereas combining the LIMK inhibitor with conventional TKIs, such as ponatinib, leads to significantly improved anti leukemic activity and longer mouse survival.

Furthermore, in in vitro model, LIMK inhibitors showed synergistic activity also with conventional chemotherapy such as dexamethasone and vincristine.

Importantly, no added toxicity was observed in CD34+ from healthy donors.

Overall, the paper is well written and the experiments are well described and results are convincing. I have no particular concerns.

The evidence that combination treatment may overcome TKI resistance in Ph pos B ALL is of interest as Loss of response to TKI is the major cause of treatment failure in this setting.

Minor

Genes should be italicized (i.e. FLT3 and not FLT3)

Author Response

Reviewer 1:

In this paper, the Authors report on the synergistic effect of LIMK inhibitors with TKIs in Philadelphia positive B cell ALL.

They show in a preclinical in vitro and in vivo model that the single agent treatment with LIMK inhibitor shows only modest activity, whereas combining the LIMK inhibitor with conventional TKIs, such as ponatinib, leads to significantly improved anti leukemic activity and longer mouse survival.

Furthermore, in in vitro model, LIMK inhibitors showed synergistic activity also with conventional chemotherapy such as dexamethasone and vincristine.

Importantly, no added toxicity was observed in CD34+ from healthy donors.

Overall, the paper is well written and the experiments are well described and results are convincing. I have no particular concerns.

The evidence that combination treatment may overcome TKI resistance in Ph pos B ALL is of interest as Loss of response to TKI is the major cause of treatment failure in this setting.

Minor

Genes should be italicized (i.e. FLT3 and not FLT3)”

Our response: We warmly thank the reviewer for the positive evaluation of our work. We have italicized genes and gene fusions mentioned in the manuscript as requested.

Reviewer 2 Report

In this manuscript, Berrou and colleagues report their results on preclinical evaluation of LIMK inhibition in BCR::ABL positive acute lymphoblastic leukemia cells. The data collected and shown in this paper demonstrate the LIMKi CEL_Amide in combination with TKI has anti-leukemic activity in Ph+ ALL models in vitro and in vivo. The study represents a preclinical demonstration of possible alternative target treatment combinations in Ph+ ALL. The introduction well explain the scenario and the main points of the study; methods and results have been well explained.

According to my opinion is an interesting and well-written report; however, I would have the following minor remarks:

1.      Line 111: …… 100 IU/mL penicillin, 100g/mL streptomycin…… Please can you verify the dose of streptomycin? Is it 100 µg/ml?

2.      Lines 179 and 184:  you have to better specify the number of the mice for each group. Is 30 the total amount or is 30 the number for each group? 

3.      Figure 1a. According to my opinion, Figure 1a is not necessary. Could it be substituted for graphs to show the different IC50 in the two cell lines?

4.      Figure 1b and 3b. Comparison for statistical significance is not so clear. Can you use some bars to show the statistically significant comparisons?

5.      Figure 1c: This figure is not a high quality image so that is not so simple to read it. Can you improve it?

6.      Figure 2: it could be better to show also LIMK2 expression in healthy donors and patients.

7.      Line 293: please correct  inotuzumab ozogamicine in the form inotuzumab ozogamicin

8.      Please check all the gene symbols and put it in italics i.e. line 76 RUNX1-RUNX1T1

Author Response

Reviewer 2:

In this manuscript, Berrou and colleagues report their results on preclinical evaluation of LIMK inhibition in BCR::ABL positive acute lymphoblastic leukemia cells. The data collected and shown in this paper demonstrate the LIMKi CEL_Amide in combination with TKI has anti-leukemic activity in Ph+ ALL models in vitro and in vivo. The study represents a preclinical demonstration of possible alternative target treatment combinations in Ph+ ALL. The introduction well explain the scenario and the main points of the study; methods and results have been well explained.

According to my opinion is an interesting and well-written report; however, I would have the following minor remarks:

  1. Line 111:…… 100 IU/mL penicillin, 100g/mL streptomycin…… Please can you verify the dose of streptomycin? Is it 100 µg/ml?
  2. Lines 179 and 184: you have to better specify the number of the mice for each group. Is 30 the total amount or is 30 the number for each group? 
  3. Figure 1a. According to my opinion, Figure 1a is not necessary. Could it be substituted for graphs to show the different IC50 in the two cell lines?
  4. Figure 1b and 3b. Comparison for statistical significance is not so clear. Can you use some bars to show the statistically significant comparisons?
  5. Figure 1c:This figure is not a high quality image so that is not so simple to read it. Can you improve it?

Figure 2: it could be better to show also LIMK2 expression in healthy donors and patients.

  1. Line 293: please correct inotuzumab ozogamicine in the form inotuzumab ozogamicin
  2. Please check all the gene symbols and put it in italics i.e. line 76 RUNX1-RUNX1T1

Our response: We thank the reviewer for the positive evaluation of our study and his helpful remarks.

Formal changes were made in the text. Number of mice is 30 in total for each experiment. This has been clarified in the text. Figure 1a was modified as suggested. Statistical modifications were done for Figure 1b and 3b and Figure 1c was improved.

Concerning western blotting of healthy donors and patients: LIMK2 was blotted for healthy donors without significant detection of protein. As we could not detect LIMK2 for healthy donors and LIMK2 was not modified in cell lines we blotted only LIMK1 in patients. Unfortunately we do not have enough material to blot them again. We apologize for this missing.

Reviewer 3 Report

The paper is well written and overall the study well conducted.

Author Response

Reviewer 3:

“The paper is well written and overall the study well conducted.”

Our response: We thank the reviewer for his encouraging comments concerning our study.